# The evolution of happiness pre and peri-COVID-19: A Markov Switching Dynamic Regression Model

Stephanie Rossouw[1]*, Talita Greyling[2], Tamanna Adhikari[3,4]

**1** School of Social Science & Public Policy, Faculty of Culture and Society, Auckland University of Technology, Auckland, New Zealand, **2** School of Economics, University of Johannesburg, Johannesburg, Gauteng, South Africa, **3** School of Business Administration, Al Akhawayn University, Ifrane, Morocco, **4** School of Economics, College of Business and Economics, University of Johannesburg, Johannesburg, Gauteng, South Africa

☯ These authors contributed equally to this work.

* stephanie.rossouw@aut.ac.nz

**Data Availability Statement:** The data is publicly available from The Gross National Happiness.today Project https://gnh.today

**Funding:** The following authors received salaries from their institutions, whom were also the funders

## Abstract

Happiness levels often fluctuate from one day to the next, and an exogenous shock such as a pandemic can likely disrupt pre-existing happiness dynamics. This paper fits a Marko Switching Dynamic Regression Model (MSDR) to better understand the dynamic patterns of happiness levels before and during a pandemic. The estimated parameters from the MSDR model include each state's mean and duration, volatility and transition probabilities. Once these parameters have been estimated, we use the one-step method to predict the unobserved states' evolution over time. This gives us unique insights into the evolution of happiness. Furthermore, as maximising happiness is a policy priority, we determine the factors that can contribute to the probability of increasing happiness levels. We empirically test these models using New Zealand's daily happiness data for May 2019 –November 2020. The results show that New Zealand seems to have two regimes, an unhappy and happy regime. In 2019 the happy regime dominated; thus, the probability of being unhappy in the next time period (day) occurred less frequently, whereas the opposite is true for 2020. The higher frequency of time periods with a probability of being unhappy in 2020 mostly correspond to pandemic events. Lastly, we find the factors positively and significantly related to the probability of being happy after lockdown to be jobseeker support payments and international travel. On the other hand, lack of mobility is significantly and negatively related to the probability of being happy.

## 1. Introduction

Research related to happiness (moods, feelings and sentiment) and COVID-19 has shown that during the pandemic, peoples' happiness decrease (Greyling et al. [1, 2]; Rossouw et al. [3]) and the number of reported negative emotions increase (Brodeur et al. [4]; Sibley et al. [5]). In saying this, we are still no closer to understanding the *dynamics of happiness*. By this, we mean

of the research. 1. Stephanie Rossouw: Auckland University of Technology via the Faculty of Culture and Society. 2. Talita Greyling: University of Johannesburg via the University Research Fund. The funders had no role in study design, data collection and analysis, decision to publish, or preparation of the manuscript.

**Competing interests:** The authors have declared that no competing interests exist.

that although we know that happiness levels change per day and over time, we do not know when these switches occur between different states (regimes) of happiness, nor do we know the length of time spent in a particular state. Furthermore, we do not know whether patterns observed when switching from one state to another change due to an exogenous shock, such as a pandemic.

Previous studies (see section 2.2 for a full discussion) that investigated the changes in emotions and well-being during COVID-19 either *did not focus on happiness or evaluative mood* or used *experimental and normal linear regression* analysis. For example, Hamermesh [6] focused on subjective well-being. In contrast, Sibley et al. [5] and Every-Palmer et al. [7] focused on psychological and emotional well-being, whereas Brodeur et al. [4] and Li et al. [8] investigated mental health. Of those studies that did focus on happiness, Greyling et al. [1, 2] used estimation techniques suited to an *experimental research design* by studying the differential effect of a treatment on a 'treatment group' versus a 'control group', assuming that the "lockdown" is the treatment. Additionally, Rossouw et al. [3] used *normal linear regression analyses* and probability estimation techniques.

Given the above, the current study makes the following contributions, i) it is the first study to use a Markov Switching Dynamic Regression Model (MSDR) to investigate the dynamics of happiness, ii) the MSDR provides us with new insights into the probabilities of transitioning from one state to another, the duration of these happiness states and the volatility of the happiness states, iii) no other study, to the knowledge of the authors, has predicted the evolution of the unobserved switches in happiness including a time period with a pandemic such as an exogenous shock, and iv) using a probit model to determine those factors which might increase the probability to be happy during a pandemic. In addition, it is one of a handful of studies to use Big Data methods combined with other data collection methods in the analysis (for comparative studies, see Brodeur et al. [4]; Greyling et al. [1, 2]; Hamermesh [6]; Li et al. [8]; Rossouw et al. [3]).

We empirically test the fitting of an MSDR model to New Zealand's daily Gross National Happiness Index (GNH) time series, a real-time measure of well-being, derived from Big Data, from May 2019 to November 2020 (see section 3.2.1 for full discussion). We choose New Zealand specifically because of the government's success in curbing the spread of COVID-19. We argue that if predictions of the unobserved switches differ for the periods before and during the pandemic in New Zealand, a country that managed to control the spread of the virus, the probability of these results being robust is high. If the same test is repeated in countries with less successful curbing strategies, we believe that the happiness dynamics differences will probably be amplified.

Therefore, in this paper, we successfully fit an MSDR model to happiness data. Estimating the model provides us with the parameters necessary to understand the dynamics of happiness before and during the pandemic. Using the MSDR allows us to derive the means of the two states of happiness, the probabilities of transitioning from one state to another, the volatility experienced during this time period, and each happiness state's expected duration. Additionally, we use the one-step method to predict the unobserved states over the time period considered. This allows us to observe the changes in the dynamics of happiness from before the pandemic to thereafter. Lastly, we investigate the factors that could likely increase the probability of being happy during a pandemic using a probit model.

The rest of the paper is structured as follows. The next section contains a short discussion on the country under investigation, New Zealand, and relevant literature. Section 3 describes the data and outlines the methodology used. The results follow in section 4, while the paper concludes in section 5.

## 2. Background and relevant literature review

### 2.1 New Zealand

The country of choice for our empirical testing is New Zealand. It is an island economy surrounded by a natural border in the Southwestern Pacific Ocean with a fairly small population of 5.5 million people. When COVID-19 reached its shores, New Zealanders had an average happiness level of 7.14 for 2020 (scale ranges from 0 to 10, see section 3.2.1 for full discussion) (Greyling et al. [9]). New Zealand was in relatively good shape in terms of the economy, with a 2.3 per cent annual GDP growth rate for 2019. New Zealand was doing better in terms of the unemployment rate than most OECD countries, with a low 2.4 per cent (Statistics New Zealand [10]). It relies heavily on the tourism sector, contributing 5.8 per cent to its Gross Domestic Product (GDP) and employs approximately 7.5 per cent of the population (Statistics New Zealand [10]). Therefore, when the New Zealand government decided to 'go fast and go hard' in response to the threat of COVID-19 on its citizens' health, it was relatively easy to close their borders to any non-residents.

The first confirmed case for New Zealand was reported on 28 February 2020, and 27 days later, on 26 March, the country went into alert level 4, which brought about a complete lockdown. Under New Zealand's level 4 lockdown, people were allowed to leave their homes only for essential reasons, but mask-wearing in supermarkets were mandatory. Additionally, they were instructed to work from home. No travelling was allowed (either domestically or international), and the schools were closed. However, they were allowed to exercise outside their homes at any given time. There was very rarely a need to enforce compliance. According to the Stringency Index (Roser et al. [11]), the mean stringency for the period (1 January to 30 May) was 41.35 (the stringency index ranges from 0 to 100, with 100 being the most stringent).

On 27 April 2020, New Zealand moved to alert level 3, which loosened some restrictions. The children of New Zealand were still not allowed to go back to school. People still had to work from home where possible, and travelling between regions was still heavily restricted. However, businesses were allowed to open with the condition that there would be no customers allowed on the physical premises. This brought about a surge in 'click and collect' retail shopping. On 4 May 2020, New Zealand succeeded in beating the virus with zero new cases to be reported. By 8 June 2020, the Ministry of Health reported no more active cases of COVID-19 in New Zealand. This brought accolades worldwide, applauding the New Zealand government's ability to have eradicated COVID-19 amidst the struggles of most countries to contain the spread of the virus.

After 102-days of being COVID-19 free, a resurgence sent New Zealand back into lockdown on 12 August 2020, but this time, different levels of stringency were imposed across the country. Now, the largest city in the country, Auckland, where 33 per cent of the population resides, was placed back into level 3 lockdown while the rest of New Zealand was moved to level 2. Nineteen days later, Auckland moved to alert level 2, but it wasn't until 7 October 2020 that Aucklanders could move freely at alert level 1.

New Zealand was one of the only countries in the world where people could gather in masses to celebrate both Christmas and the New Year. At the time of writing this paper, New Zealand once again was operating without any restrictions.

### 2.2 Literature review

According to Algan et al. [12], there is an increasing demand to use well-being measures to move beyond the classical income-based approach to measuring human development and progress. GDP does not measure non-market social interactions, such as friendship, family,

happiness, moral values or the sense of purpose in life. Additionally, Bryson et al. [13] and Pie-kalkiewicz [14] state that happiness may be a determinant of economic outcomes: it increases productivity, predicts one's future income and affects labour market performance. New Zealand has embraced this move as expressed in the Treasury's Living Standards Framework (LSF) alongside the LSF Dashboard, which the Treasury developed to inform its policy advice (McLeod [15]).

With the COVID-19 pandemic, most of the studies thus far have focused on either *subjective or psychological and emotional well-being* and mandatory lockdowns. Sibley et al. [5] and Every-Palmer et al. [7] conducted cross-sectional studies in New Zealand using online *survey data* for 1003 and 2010 respondents, respectively. Their results showed a small increase in people's sense of community and trust. However, they also found an increase in anxiety/depression post-lockdown and hinted at longer-term challenges to *psychological well-being*. In terms of *emotional well-being*, Brooks et al. [16] reviewed the psychological impact of COVID-19 published in 24 studies. Brooks et al. [16] found that most studies reported negative psychological effects, including confusion, post-traumatic stress symptoms, and anger. Similarly, to Sibley et al. [5], longer-term effects were identified. The conclusions included that where quarantine was deemed necessary, governments should take every precaution to ensure that this experience has the least negative effect on people. Additionally, findings suggested that appeals to altruism by reminding the public about quarantine benefits to wider society could be favourable.

The above negative effect on mental health was also confirmed by Brodeur et al. [4], who used *Big Data* in the form of *Google Trends* data and found an increase in searches for loneliness, worry and sadness. Staying with Big Data, Hamermesh [6] also used Google Trends data to predict married people were more satisfied with life than single people in government-imposed lockdown. Li et al. [8] used Big Data in the form of Weibo, a Chinese social media platform, in their analyses. They found negative emotions such as depression, anxiety and indignation, and sensitivity to social risks increased under COVID-19, negatively impacting *mental health*. In contrast, positive emotions such as Oxford happiness (a measure for psychological well-being) and life satisfaction decreased. People were concerned more about their health and family, while less about leisure and friends.

Apart from subjective or psychological well-being, Greyling et al. [1] and Rossouw et al. [3] conducted two studies using Big Data in the form of the GNH index to investigate the *determinants of happiness before and during the first months of the government-imposed lockdown* in an extreme country case, South Africa. They identified South Africa as an extreme country since stringent lockdown regulations were enforced against a weak economy's backdrop and already low well-being levels. Additionally, they calculated South Africa's probability *to reach the mean happiness levels* of 2019, considering the subsamples before and after the lockdown was implemented. Lastly, they investigated whether *lockdown regulations in themselves caused a decrease in happiness* in South Africa.

Their results obtained from a *difference-in-difference and OLS model*, respectively, indicated that, for an extreme country case, what significantly contributes to happiness under lockdown are the factors directly linked to the implemented regulations itself. These factors can be classified as (i) social capital issues; lack of access to alcohol, concerns about schooling and increased social media usage, and (ii) economic issues; employment concerns, the threat of retrenchments and lower levels of consumption. As expected, they found that the number of daily COVID-19 cases is negatively related to happiness. Surprisingly, they also found that the stay-at-home orders were positively related to happiness after lockdown, implying that spending more time at home, without considering the other negative effects of a lockdown, increases happiness.

Of noteworthy is Greyling et al.'s [1] finding that COVID-19, proxied by new deaths per day, had an inverted U-shape relationship to happiness. At the onset of COVID-19, seemingly South Africans were positive and optimistic as the fatality rate was relatively low and recovery rates high. However, as the pandemic progressed, South Africans became more concerned, and this relationship changed and became negative, with peoples' happiness decreasing as the number of new COVID-19 deaths increased.

Furthermore, Rossouw et al. [3] found the probability of reaching the same mean happiness levels experienced in 2019, considering the two subsamples, before lockdown regulations were implemented to be 26 per cent and after that 17 per cent. Thus, lockdown likely had a happiness cost of 9 per cent. Lastly, they found that a lockdown in itself caused a decrease in happiness.

In a third study, Greyling et al. [2] used the GNH to investigate *the relationship between lockdown and happiness*. The team included their initial three diverse countries in their analyses, namely South Africa, New Zealand and Australia. These countries differ concerning their characteristics, strictness and the duration of their respective implemented lockdown regulations. Notwithstanding the aforementioned differences, the main idea was to determine whether a *lockdown is negatively associated with happiness.* Additionally, the team compared the *well-being costs of the different degrees of strictness of these countries' lockdown regulation*s. The main results, obtained from a *difference-in-difference model*, show robust evidence of a *negative relationship between the lockdown regulations and happiness*, notwithstanding the diversity in characteristics and lockdown regulations of the countries included in the sample. Furthermore, considering the lockdown's effect size, the negative association is in increasing order of the stringency of the restrictions. Thus, *South Africa suffers the largest negative effect* compared to the other two countries, New Zealand and Australia.

Given the above literature review, and as was stated in section 1, no other study, to the authors' knowledge, has attempted to study the *dynamics of happiness*, using an *MSDR*, *before and after an exogenous shock*, such as the COVID-19 pandemic.

## 3. Data and methodology

### 3.1 Data

In the analyses, we use high-frequency daily data (see section 3.2). We analyse a time period that includes the COVID-19 pandemic and the nation's two lockdowns. The first lockdown period began on 26 March 2020 and the second on 12 August 2020. As mentioned in section 2.1, with the second lockdown, only the city of Auckland was mandated to move back into level 3 while the rest of the country moved to level 2. The time period under investigation is from 11 May 2019 to 8 November 2020 (548 days). However, we consider the pandemic period to be from 26 March, the first lockdown date, to 8 November 2020 (228 days). We chose the lockdown date since the most severe impacts of the pandemic were seen after the first lockdown was announced rather than when the first COVID-19 case was confirmed.

### 3.2 Selection of variables

To select the variables included in the models, we are led by the literature and availability of high-frequency daily data. In the next section, we firstly discuss the outcome variable, daily GNH, a real-time measure of evaluative mood and secondly, the covariates included in the probit estimations.

**3.2.1 Gross National Happiness (GNH) Index.**   To measure happiness (the outcome variable in both the MSDR and the probit models), we use the Gross National Happiness Index (GNH) launched in May 2019 for New Zealand. The GNH measures New Zealand's citizens'

happiness (evaluative mood) during different economic, social and political events. To derive our high-frequency daily data, which captures happiness, we construct the GNH using Big Data by extracting a live feed of tweets from Twitter. Subsequently, each tweet is subjected to sentiment analysis. We use the Sentiment140 lexicon in our sentiment analysis which codes tweets as either positive, negative or neutral (as a robustness test, we repeat the sentiment analyses using the lexicons Syuzhet, AFINN, Bing and NRC). The sentiment is determined by identifying the tweeter's attitude towards an event using variables such as context, tone, etc., and it helps you understand an entire opinion of the text.

After each tweet has been classified, we apply a sentiment balance algorithm to derive a happiness score per hour. The scale of the happiness scores is between 0 and 10, with 5 being neutral, thus neither happy nor unhappy. The index is available live on the GNH website (https://gnh.today). For a full description of the methodology followed, read Rossouw et al. [17]. As happiness varies over the day of the week, with a Monday low and a Friday high, we smooth the time series to remove the average day of the week effect (Kelly [18]; Helliwell and Wang [19]). The benefit of using daily time series data is the ability to reveal structural breaks in happiness across time.

New Zealand has 400,600 active Twitter users, approximately 8.37 per cent of the population (Omnicore [20]. Although the number of tweets is extensive and represents a significant proportion of the population, it is not representative. However, Twitter accommodates individuals, groups of individuals, organisations and media outlets, representing a kind of disaggregated sample, thus giving access to the moods of a vast blend of Twitter users, not found in survey data.

Additionally, after analysing GNH and the tweets underpinning the index, since 2019, it seems that the GNH index gives a remarkably robust reflection of the mood of a nation. One possible shortcoming of using the GNH, determined at the national level, is its inability to account for heterogeneity in the pandemic's effects by different groups. Therefore, we caution the reader to interpret our results as the mean impact on happiness. Ultimately, this limits the conclusions we can draw on within-country samples.

Since this is the first measure of its kind, our choices to test the GNH index's robustness is limited to correlating it to time series data, reflecting the emotions related to well-being. We used three sources to acquire the time series data: survey, Google Trends and Twitter. Firstly, we correlate New Zealand's GNH index with the 'depression' and 'anxiety' variables for the country included in the *'Global behaviours and perceptions at the onset of the COVID-19 Pandemic data'* survey for the period from 1 March 2020 (OSF [21]). We find a negative and significant relationship, mostly greater than 0.5 ($r>0.5$). Therefore, it seems that the GNH index derived from Big Data and the 'depression' and 'anxiety' variables derived from survey data give similar trends, though in opposite directions.

Secondly, we correlate the GNH index with search data for the topics of 'happiness' and 'well-being' using Google Trends. These topics were previously used in the study of Brodeur et al. [4] (see section 2.2). We find the Pearson correlation coefficient to be *r = 0.4* and *r = 0.5*, respectively.

Lastly, using Twitter data, we correlate GNH with the derived emotions, 'joy' and 'fear'. To derive these emotions, we use Natural Language Processing methods to capture the specific emotion in each word of a tweet (this method differs significantly from sentiment analysis). We find that the Pearson Correlation Coefficient is *r = 0.63* if 'joy' and the GNH are considered and *r = -0.59* if 'fear' and GNH are considered. Both relationships are significant at the 99 per cent level ($p = 0.000$).

Based on these results, we believe that the GNH index is a valid measure of a nation's evaluative mood (happiness). Our analyses use a smoothed GNH time series to account for the day-

**Table 1. Descriptive statistics of the variables.**

| Variable | Mean | Std Dev. | Min | Max | N |
|---|---|---|---|---|---|
| GNH | 7.13 | 0.30 | 5.94 | 8.1 | 548 |
| Smoothed GNH | 7.13 | 0.19 | 6.48 | 7.52 | 548 |
| Daily COVID-19 Cases | 13.11 | 22.7 | 0 | 95 | 88 |
| Border crossings (Log) | 5.20 | 2.56 | 0 | 9.44 | 88 |
| Lack of mobility | -0.99 | 1.61 | -3.79 | 1.73 | 88 |
| Jobseeker Support Payments (Log) | 11.96 | 0.93 | 11.87 | 12.14 | 88 |

Source: Authors' calculations.

of-the-week effect with a Monday low and a Friday high. Table 1 includes the descriptive statistics of both the GNH and the smoothed GNH time series.

**3.2.2 Selection of covariates used in the probit estimations.** We were challenged by the limited number of covariates we could include in estimations due to the short time period (limited observations) to avoid overfitting the models.

To represent the importance of economic factors on happiness (Sacks et al. [22]), we selected:

i.  border crossings (daily arrivals and departures into New Zealand) which is a proxy for international travel (Statistics New Zealand [10]),

ii.  daily searches on Google Trends for "jobs" as a proxy for job uncertainty in the future (Brodeur et al. [4]; Simionescu and Zimmermann [23]) and

iii.  data on weekly jobseeker support payments (JSSP) to proxy benefit payments (Statistics New Zealand [10]). We impute daily figures from weekly data using a cubic spline interpolation methodology.

After performing diagnostic tests, we found that (ii) and (iii) are highly correlated, and therefore we only included JSSP, as it was the better fitting variable.

To represent factors important to happiness other than economic, we selected:

i.  lack of mobility (significant from analysis of tweets). Here we use data derived from the COVID-19 Community Mobility Reports (Google [24]). The reports show the percentage change over time by geography across different categories of places such as retail and recreation, groceries and pharmacies, parks, transit stations, workplaces, and residential. We construct a 'lack of mobility' variable using Principal Components Analysis (PCA) to use the first extracted component as the index.

ii.  number of COVID-19 cases (exogenous shock) (European Centre for Disease Prevention and Control (ECDC) [25]).

Table 1 provides the descriptive statistics of the variables included in the models.

## 3.3 Methodology

To address our research questions, we make use of two different types of econometric models. The first is the MSDR model to investigate the dynamic of happiness. The second is a probit model to determine factors that might increase the probability of happiness during a pandemic.

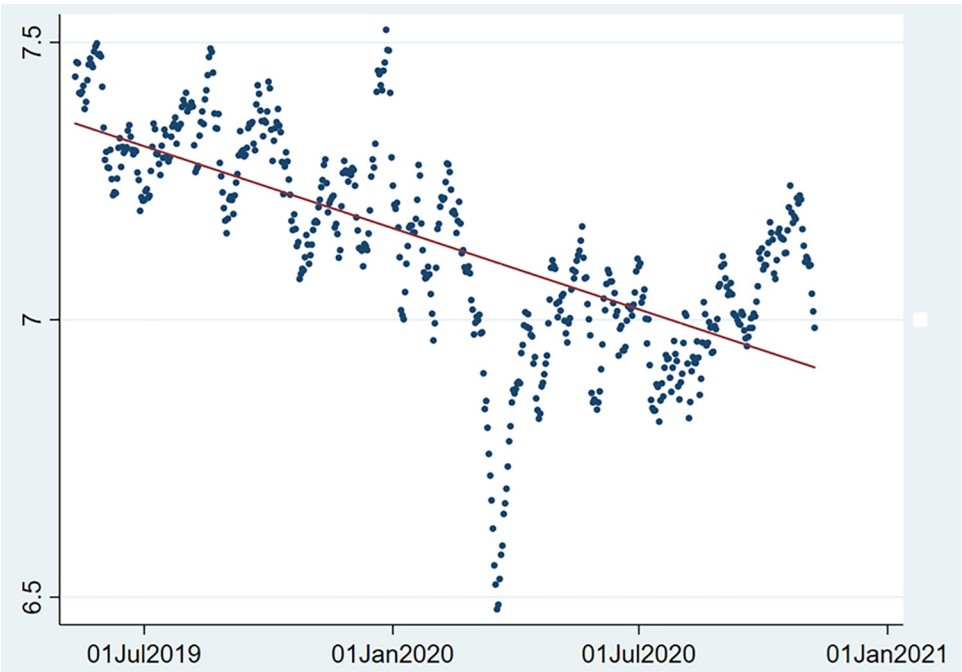

**Fig 1. Non-linearity of smoothed GNH.** Source: Greyling et al. [9].

**3.3.1 Testing for appropriateness of a non-linear model.** *3.3.1.1. Structural breaks*. We build on the framework provided by Xaba et al. [26] and Ismail and Isa [27] to test for structural breaks in the data, given that a regime-switching model is the most appropriate in the presence of structural breaks.

However, testing for structural breaks is a challenge as a priori, the timing of the break, is unknown. Therefore, we use the supremum Wald test, which uses the maximum of the sample Wald tests, to detect structural breaks with unknown timing. The test's intuition compares the maximum sample test with what could be expected under the null hypothesis of "no breaks". We find that the Wald test statistic is 512.966 ($p = 0.000$) and reject the null hypothesis of "no breaks". The estimated date for the structural break according to the Wald test is 20 March 2020. Based on this knowledge that the probability of a structural break is high, we assume that fitting an MSDR model is ideal for addressing the first research question.

*3.3.1.2 Non-linearity*. To test for non-linearity, we graphically represent the time series GNH and fit a linear line. If the plotted data points do not approximate a linear line, we can assume the data is non-linear. Fig 1 clearly shows that the smoothed GNH time series is not linear.

Next, we follow Xaba et al.'s [26] work advising to use the McLeod-Li (LM) test for non-linearity. The LM was proposed by McLeod and Li [28] based on the suggestion by Granger and Andersen [29] to test for ARCH effects. The LM test reported in Table 2 provides a formal test of evidence of ARCH effects in the GNH series.

Based on the LM test, in all the lags reported in Table 2, we reject the null hypothesis of the series being independently and identically distributed (i.i.d) (*p-value* <0 up to lag 5 reported). Thus, giving us evidence that the GNH series is possibly non-linear and dependent.

*3.3.1.3 Non-stationarity*. To test for stationarity, we use the Phillips-Perron test (Phillips and Perron [30]). The results are reported in Table 3. We can see that the test statistic of -20.77 and -12.39 (for one lag) is smaller than the corresponding critical value. This pattern is

**Table 2. LM test for non-linear dependence.**

| LM test for ARCH effects | | |
| --- | --- | --- |
| LAGS | F- Statistic | P-value |
| 1 | 510.74 | 0.00 |
| 2 | 516.63 | 0.00 |
| 3 | 515.59 | 0.00 |
| 4 | 515.04 | 0.00 |
| 5 | 514.14 | 0.00 |

Source: Authors' calculations.

repeated in the test with two lags. This indicates that we cannot reject the null hypothesis of the presence of a unit root in both cases, with and without the trend and for both lagged models. Hence, we can conclude with reasonable confidence that the GNH series is non-stationary, further indicating the unsuitability of using a traditional linear regression model. There does not seem to be any significant proof of reversion to the mean (stationarity) at the 5 per cent level of significance.

**3.3.2 The Markov-Switching Dynamic Regression Model.** To determine the dynamics of happiness, we fit a Markov-Switching Dynamic Regression Model (MSDR) to the daily GNH data for the period as described in the data section (see section 3.1).

Previously models most often employed to analyse economic variables' dynamic behaviour were linear, such as autoregressive (AR) models, moving average (MA) models, and mixed ARMA models. However, linear models have limitations; although adequate to estimate linear equations, they cannot capture non-linear dynamics such as asymmetry, amplitude dependence and volatility clustering. Furthermore, they cannot estimate models for variables that move through significant different states, such as GDP growth rates, during periods of expansion and contraction, or in the current paper, happiness levels shifting between regimes.

Therefore, non-linear models have been developed to address these shortcomings (see Granger and Terasvirta [31]). One of these models, the current analysis model, is the Markov Switching Dynamic Regression Model (MSDR) model (Hamilton [32]). It can address many of the pre-mentioned challenges. Furthermore, the model has the benefit that it can encapsulate more than one equation that characterises time series behaviours in different states, as it allows for switching between states. The MSDR's switching mechanism is controlled by an unobservable state variable that follows a first-order Markov chain. Thus, a specific structure (state) only prevails for a random time period, after which it will "switch" to another structure (state). Thus, using MSDR, we do not need to make subjective judgements on the state we are in, a priori, as the MSDR model itself determines the switch. From the model, we derive the transitioning probabilities of the happiness time series from switching from one state to another and the duration of the states of happiness.

In choosing the best fitting model, we performed estimations using different states and included covariates such as lagged GNH. Additionally, we also estimated an autoregressive

**Table 3. Tests of non-stationarity.**

| Phillips-Perron Test | Trend | Critical Value (5%) | Without Trend | Critical Value (5%) |
| --- | --- | --- | --- | --- |
| Lag(1) | -20.77 | -21.88 | -12.39 | 14.10 |
| Lag(2) | -23.06 | | -13.62 | |

Source: Authors' calculations.

Markov Switching model. Based on our diagnostic tests results and by a lower information criterion (AIC and BIC) of the MDSR compared to the autoregressive Markov Switching model, we believe the dynamic model to be appropriate as it allows for a quick adjustment after the Markov process changes state. Further, a dynamic model is appropriate for high-frequency data, such as our estimation data. To specify the process in state ($s$) at time $t$, we can write the MSDR model as:

$$GNH_t = \mu_{st} + e_{st} \tag{1}$$

Where $s$ represents the unknown states and $t = (0,1,2....)$ days.
Parameter $\mu$ is state-dependent and represent the stochastic trend.

$$E(S_{t-1}) = \mu_s \tag{2}$$

$e_{st}$ is an independent and identically distributed (i.i.d.) normal error with mean 0 and variance $\sigma_s$ which is state-dependent.

We further assume that the Markov chain $s$ is ergodic with transition probabilities between states given by:

$$P_{ij} = P(S_t = i) \tag{3}$$

$$where \; i, j = (1, 2)$$

The set of six parameters (two transition probabilities, two trends and two volatility parameters) are estimated using an Expectation Maximisation algorithm. As the number of parameters estimated increases rapidly as the number of states increases and given the relatively small number of observations for our sample, we tested for the minimum of 2 states, which seems appropriate. We report robust standard errors to correct for heteroscedasticity inherent to high-frequency data.

The classification of the states is largely subjective and can be inferred upon observing the state-dependent $\mu$. In this way, it seems that the two states identified can be classified as an *unhappy* and *happy* state. The naming convention of "unhappy" and "happy" state is to distinguish between the regimes of happiness in New Zealand and no indication of happiness levels relative to other countries.

Finally, given the estimated transition probabilities, we compute the expected duration (D) in each of the two states through the following equation:

$$E(D_S) = \frac{1}{1 - P_{ij}} \tag{4}$$

$$where \; i, j = (1, 2)$$

Once we have fitted the MSDR model, we use the estimated parameters to predict the evolution of the states over time. In other words, the state happiness will be in for the next day spanning the period May 2019 –November 2020 using the previous information on the dependent variable (all the data on the previous days–thus smaller than $t$ (current day)). The non-linear filter is performed on previous periods using the one-step method, but only one-step predictions are made for the current period. Therefore, the one-step predictions are the forecasted values of the dependent variable using one-step-ahead predicted probabilities.

**3.3.3 Probit regression.** To address our second objective to determine which factors could likely increase the probability to be happy during the pandemic (26 March 2020 to 8 November 2020), we make use of a probit model with 1 (GNH $\geq$ 7.13) (Happy > Unhappy

state) the happy state and 0 (GNH< 7.13) (Happy < Unhappy state) the unhappy state.

$$Pr(GNH \geq 7.13) = \alpha_0 + \alpha_2 X_t + \mu_t \qquad (5)$$

We include a vector of covariates encapsulated in $X_t$ (see section 3.2.2) with $\mu_t$ the error term. $X_t$ includes border arrivals, the number of people seeking jobseeker support payments, lack of mobility, and lagged daily new COVID-19 cases.

In the absence of panel data or an appropriate instrument, it is difficult to assess the likelihood of endogeneity. However, by performing a basic test of correlating the error term with the covariates, we find the correlation between each covariate and the error term weak. Additionally, our results are in line with expectations, both considering the sign and significance of covariates; therefore, we believe our results are a good reflection of the correlations between our dependent variable and the covariates. Lastly, we like to note that we do not claim any causality in these estimations. With that being said, we are conscious of these limitations.

## 4. Results and analysis

This section reports on the empirical results obtained from the MSDR and probit models.

### 4.1 Results of the MDSR

We use the MDSR with a two-state regime to extract the happiness states for New Zealand, spanning the period 11 May 2019 to 08 November 2020. Table 4 provides the estimated parameters, which describe the two states of happiness. We define the two states as a '*happy state*' with a mean happiness (GNH) level of 7.30 and an '*unhappy state*' with a mean happiness (GNH) level of 6.97. The two sigmas reported in Table 4 are state-dependent and indicate the volatility in the respective states. The volatility in the happy state is 0.142 and the unhappy state 0.176. Using the variance comparison test, we find that the difference in the variance in the two states is significant at the 95 per cent level. We note that the volatility in the unhappy state is relatively higher than in the happy state. This means that when people are in the unhappy state, the volatility of happiness (GNH) increases, reflecting higher highs and lower lows. A likely explanation for this can be higher levels of emotions (positive and negative) in times of uncertainty. Lower levels of happiness can be due to the announcement of a lockdown. In contrast, restoring certain liberties previously taken away by strict lockdown regulations can explain higher happiness levels.

Table 5 contains the probabilities of switching from one state (either happy or unhappy) to another in one period of time (one day) to the next period (the next day) and only depends on the previous state.

From the results in Table 5, it seems that both states are persistent. If New Zealanders are in an unhappy state, the probability to stay in the unhappy state is 0.966. The same holds if the

**Table 4. Markov switching model parameters for the period 2019–2020.**

| Smoothed GNH | Unhappy State | Happy State |
| --- | --- | --- |
| Mean | 6.966 (0.015) *** | 7.301 (0.012) *** |
| Sigma | 0.176 (0.009) | 0.142 (0.007) |
| p11 | 0.966 (0.013) | |
| p21 | 0.033 (0.011) | |
| N | 548 | |

Source: Authors' calculations. Standard errors in parenthesis

**Table 5. Transition probabilities of switching between the happy and unhappy state.**

|  | Unhappy state | Happy state |
|---|---|---|
| Unhappy state | 0.966 (0.013) | 0.034 (0.013) |
| Happy state | 0.033 (0.011) | 0.967 (0.011) |

Source: Authors' calculations.

country is happy, with the probability of staying in a happy state being 0.967. Seeing that the columns' conditional probabilities sum to one, it is clear that transitioning to the alternative state is very low, from unhappy to happy is only 0.034 and from happy to unhappy only 0.033.

Table 6 shows the average duration to stay in a specific state. The average length to stay in the unhappy state is 31 days, whereas to stay in the happy state is 30 days. According to these results, New Zealanders stay in the unhappy and happy state for similar time periods. Therefore if New Zealand is in an unhappy state, affirmative policy intervention can switch the country to a happy state with a probable duration of 30 days.

Fig 2 shows the smoothed GNH and the one-step predicted states. The one-step predicted states predict the evolution of the unhappy state (state 1) over time and assist with interpreting the effect of the pandemic on the happiness states.

Let's consider the probabilities to be unhappy over time. It is clear that in 2019 the probabilities are mostly close to zero, whereas, in 2020, there are distinct time periods when the probabilities of being unhappy are close to one. Since the beginning of 2020, we have distinguished four periods where the probability of being unhappy is close to one. Of those, three is related to the pandemic. In early January 2020, the first was not related to the pandemic but instead to the natural disasters in their neighbouring country, Australia. As 11 per cent of New Zealanders call Australia their home, New Zealand's closest ally suffered devastating bushfires causing New Zealand to dispatch its defence forces to mitigate the seemingly unstoppable disaster. The second time period corresponds to when Australia reported their first confirmed COVID-19 case on 25 January 2020 located in Melbourne, which brought the international pandemic and the associated risks for New Zealanders significantly closer to home.

The third and fourth time periods coincide with the first and second lockdowns. As discussed in section 2.1, New Zealand entered its strictest level (level 4) of lockdown on 26 March 2020 and only returned to level 3 on 27 April 2020. After eradicating COVID-19 in May 2020, four new community cases were reported on 11 August 2020. The Prime Minister, Jacinda Ardern, a staunch believer in going '*hard and fast*', placed 33 per cent of the population (city of Auckland) back into level 3 lockdown on 12 August 2020. The rest of New Zealand was moved to level 2. Nineteen days later, Auckland moved to alert level 2, but it wasn't until 7 October 2020 that Aucklanders could move freely at alert level 1.

From these results, it is clear that considering the evolution of being unhappy over time, the unhappy state was dominant in 2020. Considering these results and that of Table 6, which showed that staying in the unhappy state has an expected duration of 48.67 days, it emphasises the need for affirmative policy intervention during unhappy states to improve well-being.

**Table 6. Expected duration in each state (in days) for May 2019 –November 2020.**

|  | Unhappy state | Happy state |
|---|---|---|
| Mean Duration | 31.00 | 30.11 |
| Standard Error | 12.47 | 10.24 |

Source: Authors' calculations.

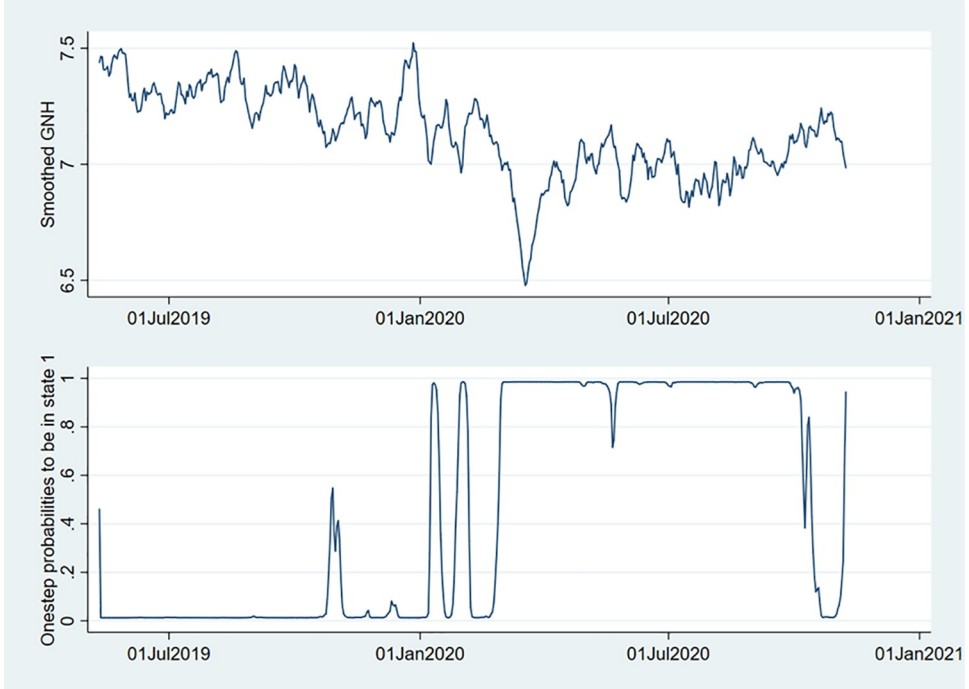

**Fig 2. Smoothed GNH and one-step predictions of being in an unhappy state.** Source: Greyling et al. [9].

Having identified a persistent unhappy state issue, we now move to determine those factors that can influence the probability of happiness. This could serve policymakers with information on how to return to pre-pandemic happiness levels.

## 4.2 Probit results

We use a probit model to estimate the probability to be happy in a time period of a pandemic, with restrictions in place to curb the spread of COVID-19. The factors significantly related to happiness after the first lockdown was announced are jobseeker support payments, border crossings, and lack of mobility (see Table 7).

We note that international travel (border crossings) increases the probability to be happy. At the time of writing this paper, New Zealand was still operating under a 'closed border' policy, which has a double impact on New Zealanders' happiness. First, as an economic shock and second as a social shock. Those impacted directly by the lack of international and domestic tourism experience a significant economic shock that negatively influences their livelihoods. According to Statistics New Zealand [33], overseas visitor arrivals were down by 405 300 to 5400 in January 2021, compared with January 2020. In the ten months from April 2020 to January 2021, there were a mere 42800 visitor arrivals, down 3.1 million compared with the same ten months from April 2019 to January 2020, leaving a 6-billion-dollar shortfall (Statistic New Zealand [33]).

Additionally, New Zealanders are known for their international travelling. In the year 2019, over 1.6 million New Zealanders (35.21 per cent of the total population) travelled abroad (Statistics New Zealand [33]). Not being able to go and travel the world is a social shock causing a decrease in happiness. At the time of writing this paper, New Zealand had no new COVID-19 community transmissions. Although Australia and the Pacific Islands were earnestly looking to secure a way to open up tourism travel with New Zealand, the government was still not ready to commit.

**Table 7. Probit estimations: Factors likely to increase the probability to be happy during the pandemic.**

| | Full sample | |
|---|---|---|
| | **Coefficient** | **SE** |
| COVID-19 Cases | -0.016 | (0.009) |
| Border cross (arrivals) | 0.0001*** | (0.00003) |
| Jobseeker support (log) | 0.00004*** | (0.00001) |
| Lack of mobility | -0.178** | (0.091) |
| Cons | -8.8401*** | (2.625) |
| N | 300 | (0.0152) |

Source: Authors' calculations. Note: to account for heteroscedasticity, we report robust standard errors in parentheses.

Note: Lack of mobility refers to restrictions on local travel while border crossing captures international travel.

* $p < 0.10$

** $p < 0.05$

*** $p < 0.01$.

Jobseeker support payment (JSSP) also increase the probability to be happy. JSSP is a temporary benefit paid for up to 52 weeks while unemployed New Zealanders look for employment opportunities, find themselves in training for work or are unable to work due to health conditions, injury or disability (Ministry of Social Development [34]). The benefit amount ranges from, for example, a single person 18–19 years of age living at home receiving $175.48 per week, whereas a solo parent could receive up to $375.17 per week. Comparing the years 2019 to 2020, we saw a significant increase of 43.4 per cent in the number of jobseeker support applicants (Statistics New Zealand [33]).

On the other hand, lack of mobility decreases the probability to be happy. As discussed in section 2.1, while New Zealanders understood that measures had to be implemented to curb the spread of the virus, the limited mobility, being forced to work from home, children not being allowed to attend schools, and only leaving their homes for essential reasons had a negative effect on happiness.

We want to remind the reader that the probit model is a correlation exercise that could shed some light on those factors associated with the probability of happiness for New Zealand. Admittedly, the list of covariates is limited due to the limited availability of high-frequency daily data. However, we believe that despite these methodological limitations, the results can help us draw important conclusions.

## 5. Conclusions

This paper successfully fitted a Markov Switching Dynamic Regression Model (MSDR) to better understand the dynamic patterns of happiness levels. Furthermore, we investigated how the happiness patterns change over time, including a time period characterised by an exogenous shock, namely the pandemic. Lastly, we determined the factors that will contribute to happiness levels during the time of a pandemic.

This study is unique as it is the first study to use MSDR to investigate the dynamics of happiness and its evolution over time. The time period under investigation is also unique as it includes an unprecedented exogenous shock, namely the COVID-19 pandemic. Additionally, it is one of the few studies to combine Big Data methods with other data to derive a dataset for analysing daily happiness.

The MSDR in the study showed that New Zealand switches between two states of happiness: a happy (lower volatility) and an unhappy (higher volatility) state, and both states are persistent. This means that once New Zealanders are in a happy or unhappy state, the probability of moving out of these states is very low. Furthermore, considering the one-step prediction probabilities of being in an unhappy state to estimate the evolution over time, we find that relative to a time period before the pandemic, the frequency of the probability of being unhappy the next day is very high. In the year of the pandemic, we identified four periods where the probabilities of being unhappy were high. Three out of the four time periods were directly linked to events related to the pandemic. Such as the announcements and implementations of lockdowns.

Lastly, we found the factors positively and significantly related to the probability of being happy after lockdown to be jobseeker support payments and international travel (border crossings). On the other hand, lack of mobility is significantly and negatively related to the probability of being happy.

For happiness levels to return to pre-pandemic levels, policymakers could intervene by not imposing any other unjustified lockdowns such as those seen in 2021. The lockdowns of 2020 already had a severe negative impact on New Zealanders' happiness levels, yet the government overreacted twice in the year 2021. For example, following the New Zealand governments strategy of 'going hard and fast', Auckland was again sent back into lockdown level 3 for three days with a mere three new community transmission cases on 14 February 2021. Auckland was kept in lockdown until 22 February despite no new daily positive cases reported after the 14[th]. Seeing that 28 per cent of the Auckland workforce cannot operate under level 3, the estimated cost to the economy for the period spent in level 3 lockdown was a massive $60-$69 million (ASB [35]). On 28 February 2021, Auckland was sent back into level 3 lockdown for one week because of only one case of COVID-19. Again, despite no new daily positive cases being reported, the lockdown was not lifted sooner. This time the cost to the economy was a staggering $240 million (ASB [36]). Additionally, New Zealanders' happiness levels fell to 6.72 on 28 February, with many people expressing '*that they cannot do this again*'. Subsequently, these lockdowns did not bring the previous accolades. Instead, the world reacted the same as New Zealanders, asking how the government could justify stripping people of their basic rights and costing the country millions of dollars, all for the sake of only one case of COVID-19.

Additionally, the government should open the borders to allow international travel. This could restore mobility and encourage border crossings, thereby increasing tourism to rekindle the economy. The possibility of establishing a 'Trans-Tasman bubble' to encourage international travelling and tourism could be one of the possible areas for the government to focus on. The extended bubble could go further than just Australia and include other low-risk nations such as Singapore. Additionally, 21 island nations and territories in Oceania have reported no COVID-19 cases, including Samoa, American Samoa, Tonga, Tuvalu, Tokelau, Niue, Nauru, Kiribati, the Cook Islands and the Solomon Islands. In saying this, the government should only consider this option without increasing the COVID-19 risk again. Additionally, the government could also create employment opportunities by focusing on the construction sector (since New Zealand has a housing shortage) to decrease the fear of losing jobs and the dependency on jobseekers' support. Failure to increase happiness levels could have further negative spill-over effects in various economic, social, and political domains.

## Acknowledgments

We would like to thank our colleagues Professor Emeritus Philip S Morrison from the Victoria University of Wellington and Conal Smith from Kōtātā Insight, for their generosity in providing feedback on the study. Additionally, we thank AFSTEREO for the I.T. support provided.

## Author Contributions

**Conceptualization:** Stephanie Rossouw, Talita Greyling, Tamanna Adhikari.

**Formal analysis:** Stephanie Rossouw, Talita Greyling, Tamanna Adhikari.

**Investigation:** Stephanie Rossouw, Talita Greyling, Tamanna Adhikari.

**Methodology:** Talita Greyling, Tamanna Adhikari.

**Resources:** Stephanie Rossouw, Talita Greyling, Tamanna Adhikari.

**Software:** Talita Greyling, Tamanna Adhikari.

**Supervision:** Stephanie Rossouw.

**Validation:** Talita Greyling, Tamanna Adhikari.

**Visualization:** Stephanie Rossouw.

**Writing – original draft:** Stephanie Rossouw, Talita Greyling, Tamanna Adhikari.

**Writing – review & editing:** Stephanie Rossouw, Talita Greyling, Tamanna Adhikari.

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
