## [Decision Letter · Decision Letter 0]

29 Jul 2021

PONE-D-21-12326

The implied volatility of happiness pre and peri-COVID-19: a Markov Switching Dynamic Regression Model.

PLOS ONE

Dear Stephanie Rossouw,

Thank you for submitting your manuscript to PLOS ONE. After careful consideration, we feel that it has merit but does not fully meet PLOS ONE’s publication criteria as it currently stands. Therefore, we invite you to submit a revised version of the manuscript that addresses the points raised during the review process.

We look forward to receiving your revised manuscript.

Kind regards,

Maximo Rossi, PhD Economics

Academic Editor

PLOS ONE

Additional Editor Comments (if provided):

Dear Dr. Stephanie Rossouw,

thank you for submitting your manuscript to PLOS ONE. After careful consideration, we feel that it has merit but does not fully meet PLOS ONE’s publication criteria as it currently stands. As you will see, the referees found your proposal interesting but made a number of important observations about the current version. If you are interested in considering the observations of the referees, we invite to submit a revised version of the manuscript that addresses all the the points raised during this review process. If you decide that, please submit your revised manuscript by October 31, 2021. If you will need more time than this to complete your revisions, please reply to this message or contact the journal office at plosone@plos.org.

Best, Maximo Rossi

Journal Requirements:

NO

Reviewers' comments:

Reviewer's Responses to Questions

**Comments to the Author**

1. Is the manuscript technically sound, and do the data support the conclusions?

Reviewer #1: Partly

Reviewer #2: Yes

2. Has the statistical analysis been performed appropriately and rigorously? 

Reviewer #1: I Don't Know

Reviewer #2: Yes

3. Have the authors made all data underlying the findings in their manuscript fully available?

Reviewer #1: Yes

Reviewer #2: Yes

4. Is the manuscript presented in an intelligible fashion and written in standard English?

Reviewer #1: Yes

Reviewer #2: Yes

5. Review Comments to the Author

Reviewer #1: The article addresses a very important topic when considering well-being during a pandemic. The authors do an impressive job dealing with big data. The paper is clearly written and well-organized.

Major comments

1. First, as one of the strengths of the paper is the use of big data -GNH- it deserves a comprehensive explanation. It is not possible to follow an understand the paper without it.

2. I find the two methods applied as non-connected. I couldn’t understand why they are both applied to the data in the same paper.

3. Probit estimates are uninformative. As it is said by the authors, there are serious endogeneity problems, and the set of variables chosen is too narrow. As these results have serious problems, I recommend the authors completely re-think the estimation or even consider avoiding them and concentrate on the MSDRM.

4. As the probit estimates are weak, conclusion shouldn´t be based on them.The authors have to be careful in concluding based on them.

5. The variable “mobility” is confusing. At some point it is explained that it is actually lack of mobility, but all along the paper the authors refer to it as simply mobility and addressing its negative relation to happiness, what is confusing.

Minor comments

1. Literature review is too long, maybe it would be better to have a shorter an focused on COVID section.

2. Line 107- “average happiness level of 7.14”. As GNH is not explained before, I don’t know what that means.

3. Line 545-546- the authors refer to “unjustified lockdowns” and that the “government overreacted”. I that has to be explained in the context of a country that successfully managed the pandemic.

Reviewer #2: Authors study the impact of COVID-19 pandemic on daily happiness levels for New Zealand from May 2019 to September 2020. They evaluate the probability of switching happiness states, the duration of each state and the impact of different factors (mobility, travel, lockdowns) on happiness. Finally, they conclude suggesting different affirmative policies to minimize the negative impact of the pandemic.

My comments relate to fundamental intuitions underlying the study.

First, authors consider two states of happiness: a "happy state" and an "Unhappy state". Although I understand the objective, in reality there are no two such states since New Zealand has historically showed high levels of happiness. New Zealanders are happy people. I think authors should at least mention this fact.

Second, authors begin stating that levels of happiness change every day and are very volatile. This is NOT a fact in New Zealand during the period analyzed: not only happiness levels have remained high (according to the data from GNH) but have shown low volatility (their Tables 1 and 4). I consider this an important weakness of the paper: an attempt to explain changes in a dependent variable that have not changed much during the pandemic.

Third, results and conclusions show what one would assume as normal during a pandemic and reflect the trade-off between the economic and health and social outcomes. This trade-off has been at the core of the debates worldwide. Moreover, one would assume that banning international travel in an essentially two-way tourist country such as NZ would negatively impact on happiness level regardless of a pandemic.

The comment above is related to omitted variables. I would suggest authors to include (if possible) variables measuring satisfaction with government during the period, for example. Also, lack of mobility and border crossings may be highly correlated which implies that including both variables in the same regression could bias the results. I suggest checking this and maybe look for interactions.

On recommendations, authors suggest opening international travel, easing lockdown measures and mobility. But this would probably increase the risk of new COVID cases, so authors do not address the possible impact of this trade-off on happiness, which would make the paper much more interesting than it is in its current version (which is an interesting paper already!). Authors stress the negative impact of the Government overreacting twice during 2021 but do not explain why the government made that decision and the precise impact it had on happiness levels.

6. PLOS authors have the option to publish the peer review history of their article (what does this mean?). If published, this will include your full peer review and any attached files.

Reviewer #1: No

Reviewer #2: No

---

## [Author Response · Author response to Decision Letter 0]

4 Oct 2021

Reviewer comments addressed by the authors 

Reviewer #1: 

The article addresses a very important topic when considering well-being during a pandemic. The authors do an impressive job dealing with big data. The paper is clearly written and well-organized.

We thank the reviewer for this kind and generous review of our study.

Major comments

1. First, as one of the strengths of the paper is the use of big data -GNH- it deserves a comprehensive explanation. It is not possible to follow an understand the paper without it.

We thank the reviewer for this comment. Subsequently, we have elaborated on how the GNH is calculated. Please see section 3.2.1 (lines 225 – 238).

2. I find the two methods applied as non-connected. I couldn't understand why they are both applied to the data in the same paper.

We thank the reviewer for this comment regarding the connection between the two methods. The purpose of the MSDR model is to investigate the dynamics of happiness. It provides us with new insights into the probabilities of transitioning from one state to another, the duration of these happiness states and the volatility of the happiness states. Additionally, it predicts the evolution of the unobserved switches in happiness during a time period that includes an exogenous shock (the pandemic). Therefore, it is the method that identifies (diagnoses) the problems that need policy intervention. The MSDR signals that New Zealanders have fallen into a state of reduced happiness levels and that such a state is quite persistent. 

We then turn to the probit model, seeing as this method helps us provide policy advice by identifying variables significantly related to the probability of being in a happy state. Policymakers can address these factors that are identified to increase happiness levels in New Zealand.

We have added a further explanation in the paper reflecting the above. Please see lines 77, 95, 96, 298 – 300 and 404 – 405. 

3. Probit estimates are uninformative. As it is said by the authors, there are serious endogeneity problems, and the set of variables chosen is too narrow. As these results have serious problems, I recommend the authors completely re-think the estimation or even consider avoiding them and concentrate on the MSDRM.

We thank the reviewer for this comment. However, we believe in this paper the probit estimation is informative as it indicates the factors that policymakers can address to increase the probability to be happy in New Zealand. We want to emphasise that we did not categorically state that there were endogeneity problems; we merely, out of an excess of caution, warned that there might be. Subsequently, we used diagnostic tests for endogeneity. We found the error term and the independent variables are correlated but so slightly that they should not affect the significance of the covariates. We ran an OLS with happiness as a continuous variable to further test our findings and found the same results. Therefore, we trust our results are robust and can be informative to policymakers. Furthermore, we remind the reviewer that we make no claims of causality; however, to make it clearer to the reader, we rephrased this section. Please see section 3.3.3, lines 412 – 418. 

Regarding our chosen variables, we want to highlight that the set of covariates that have been chosen was dictated by literature and data availability as we needed real-time high-frequency daily data. Additionally, no model can include all covariates; therefore, the error term captures any unexplained variance. Thus, the limited number of covariates does not reflect a weak model.

4. As the probit estimates are weak, conclusion shouldn't be based on them. The authors have to be careful in concluding based on them.

We thank the reviewer for this comment. We agree with the reviewer regarding the model's limitations, though we do not believe the results are weak. We believe that the results are robust. However, we have included a caveat regarding our results stating the same. Please see lines 532 – 535 on page 22.

5. The variable "mobility" is confusing. At some point it is explained that it is actually lack of mobility, but all along the paper the authors refer to it as simply mobility and addressing its negative relation to happiness, what is confusing.

We thank the reviewer for pointing out this oversight. Subsequently, 'mobility' has been changed to reflect 'lack of mobility' throughout the revised manuscript.

Minor comments

1. Literature review is too long, maybe it would be better to have a shorter an focused on COVID section.

We thank the reviewer for this comment. Subsequently, the literature review has been shortened. However, we wish to point out that the other studies' detailed discussions using the GNH remain. We believe this is necessary to clarify our study's contributions and how our study differs from theirs. 

2. Line 107- "average happiness level of 7.14". As GNH is not explained before, I don't know what that means.

We thank the reviewer for this comment. We have added to the GNH discussion following comment #1 and believe it is appropriate to refer the reader here to section 3.1.2 for the complete discussion in lines 105 and 106. We also added a scale ranging from 0 to 10 to allow better context of the score of 7.14.

3. Line 545-546- the authors refer to "unjustified lockdowns" and that the "government overreacted". I that has to be explained in the context of a country that successfully managed the pandemic.

We thank the reviewer for this comment. Subsequently, we expanded the discussion to ensure the reader understands that the two lockdowns discussed were not justified given the small number of positive cases (three and then one). Given that no new daily cases were reported in the following days, the government should have lifted the restrictions sooner to minimise the cost on the economy and New Zealanders' well-being. Please see lines 557 – 571 on page 23.

Reviewer #2:

Authors study the impact of COVID-19 pandemic on daily happiness levels for New Zealand from May 2019 to September 2020. They evaluate the probability of switching happiness states, the duration of each state and the impact of different factors (mobility, travel, lockdowns) on happiness. Finally, they conclude suggesting different affirmative policies to minimize the negative impact of the pandemic.

My comments relate to fundamental intuitions underlying the study.

1. First, authors consider two states of happiness: a "happy state" and an "Unhappy state". Although I understand the objective, in reality there are no two such states since New Zealand has historically showed high levels of happiness. New Zealanders are happy people. I think authors should at least mention this fact.

We thank the reviewer for this important comment. However, the MSDR distinguishes between two states. To differentiate between the two states, we named the state with a relatively lower mean happiness level "unhappy" and the state with a relatively higher happiness level "happy". However, this is only a naming convention and does not reflect the levels of happiness relative to other countries. We added footnote 2 on page 16 to explain the naming convention.

Furthermore, we understand that this issue of happiness levels warrants further explanation. New Zealanders indeed have, on average, higher levels of happiness than other countries for which a similar GNH analysis can be conducted. However, we believe that trends in intra-country happiness levels over time are also important to investigate. The fact remains that happiness levels in NZ have reduced relative to its 2019 levels. We believe this should be a cause of concern for policymakers, particularly as we now have literature showing negative effects of reduced happiness on other economic indicators like GDP, growth rates, and unemployment rates (Bryson et al. 2016 and Piekalkiewicz 2017). Subsequently, we explained in the manuscript that the names chosen for the two states are subjectively chosen to differentiate between the states. However, these states reflect comparisons within New Zealand and not between New Zealand and other countries.

References

Bryson, A., Clark, A. E., Freeman, R. B., & Green, C. (2016). Share capitalism and worker well-being. Labour Econ, 42: 151-158.

Piekalkiewicz, M. (2017). Why do economists study happiness? Labour, 28(3): 361-377.

2. Second, authors begin stating that levels of happiness change every day and are very volatile. This is NOT a fact in New Zealand during the period analyzed: not only happiness levels have remained high (according to the data from GNH) but have shown low volatility (their Tables 1 and 4). I consider this an important weakness of the paper: an attempt to explain changes in a dependent variable that have not changed much during the pandemic.

We thank the reviewer for this observation and realised that "volatility" and definitions of volatility needs further explanation.

Volatility is defined as the up and down movement of stock markets or the prices of stocks. In our paper, we do not have stock markets or prices but the GNH. There can be high volatility (big swings) or low volatility (smaller changes) in the prices of stocks (GNH) (Green 2021). Although different definitions are used to determine high and low volatility in financial markets, the fact remains that when the prices of stocks and stock markets move up and down, they are defined as volatile.

See, for example, the following definition and explanation from Boyte-White (2021), "The simplest definition of volatility is a reflection of the degree to which price moves. A stock with a price that fluctuates wildly—hits new highs and lows or moves erratically—is considered highly volatile. A stock that maintains a relatively stable price has low volatility." Additionally, we refer you to the work done by Wolf (2005). 

There are certain rules of thumb regarding whether stock prices are deemed to have high or low volatility. For example, if the standard deviation is greater than one, the stock price is seen as highly volatile and less than one less volatile. Additionally, if the percentage change in the price of a stock is more than one per cent, it is seen as more volatile (Hayes 2021). 

These rules of thumb apply to financial markets. As the current paper is the first of its kind, there is no pre-existing rule of thumb for happiness or subjective well-being levels. However, applying the financial guidelines and rules of thumb, we are confident that GNH is volatile. The degree of volatility can only be determined if it can be compared to the findings of other similar papers (there are none). Or if we compare the volatility between states. Therefore, we extend our analysis to include this comparison; see section 4.1 and Table 4 on pages 17 and 18. We also added the descriptive statistics of the GNH before smoothing the time series, see Table 1 – which gives a better indication of the volatility of the data. 

Considering the definition of high volatility in financial markets, as discussed above, the GNH is volatile and changes daily – it even changes hourly (please see www.gnh.today). For a visual presentation of volatility, figure 1 in the response document shows the daily percentage change in the GNH over the time period under investigation. The most significant percentage increase in the GNH was 17 per cent (13 August 2020), and the most significant decrease was 12 per cent (11 June 2020). Therefore, according to the rule of thumb of a one percentage change per day, the GNH is indeed volatile.

The problem with volatile data is fitting an efficient model for parameter estimations and predictions. This is why the MSDR model is often used in Financial Economics, as the data, for example, on stocks, interest rates and forex, are volatile. 

The volatility of the GNH is also why we fitted an MSDR model to ensure a better fit and robust estimation of parameters. The second benefit of the MSDR is that it can highlight a structural break in data without prior specification. In our instance, the MSDR indicates a break in the data corresponding with the lockdown, and therefore the differences in means and standard deviations of the two regimes (see Table 4). 

From figure 2 in the response document which depicts the GNH over time, the volatility is also apparent.

To your comment: "I consider this an important weakness of the paper: an attempt to explain changes in a dependent variable that have not changed much during the pandemic".

Please see the explanations above and also note the percentage change in the GNH for the two different regimes. Therefore, the dependent variable did show considerable change over the period.

1) Happy regime: the decrease in the GNH for the happy regime was from 7.5 to 6.48, which is a 12.2 per cent change.

2) Unhappy regime: from 6.46 to 7.05, which is a 9 per cent change.

Lastly, we wish to inform the reviewer that we decided to change the paper's title since we do not wish to create any confusion among readers. The new title is now "The evolution of happiness pre and peri-COVID-19: a Markov Switching Dynamic Regression Model", and we trust the reviewer will find this change agreeable.

References

Boyte-White, C. (2021). What Is the Best Measure of Stock Price Volatility? Available online https://www.investopedia.com/ask/answers/021015/what-best-measure-given-stocks-volatility.asp

Green, T. (2021). Stock Market Volatility Defined. Available online https://www.fool.com/investing/how-to-invest/stocks/stock-market-volatility/

Hayes, A. (2021). Volatility. Available online https://www.investopedia.com/terms/v/volatility.asp

Wolf, H. (2005). Volatility: definitions and consequences. In J, Aizenman and B. Pinto Managing economic volatility and crises: A practitioner's guide, p.45-64.

3. Third, results and conclusions show what one would assume as normal during a pandemic and reflect the trade-off between the economic and health and social outcomes. This trade-off has been at the core of the debates worldwide. Moreover, one would assume that banning international travel in an essentially two-way tourist country such as NZ would negatively impact on happiness level regardless of a pandemic.

The comment above is related to omitted variables. I would suggest authors to include (if possible) variables measuring satisfaction with government during the period, for example. Also, lack of mobility and border crossings may be highly correlated which implies that including both variables in the same regression could bias the results. I suggest checking this and maybe look for interactions.

We thank the reviewer for this comment.

i) We would have appreciated the opportunity to add real-time high-frequency daily data on satisfaction with the government, but unfortunately, it does not exist. However, we do agree the results on such a variable would have been insightful. Please see lines 272 – 273 on page 10, stating the same.

ii) Lack of mobility is domestic in nature and reflects the percentage change over time by geography across different categories of places such as retail and recreation, groceries and pharmacies, parks, transit stations, workplaces, and residential. On the other hand, border crossings refer to international travel. The two capture different aspects of mobility and are only moderately correlated (correlation coefficient 0.36). We do thank the reviewer for this comment and can understand why it can be confusing. We have thus added a footnote under Table 7 explaining this distinction.

4. On recommendations, authors suggest opening international travel, easing lockdown measures and mobility. But this would probably increase the risk of new COVID cases, so authors do not address the possible impact of this trade-off on happiness, which would make the paper much more interesting than it is in its current version (which is an interesting paper already!). Authors stress the negative impact of the Government overreacting twice during 2021 but do not explain why the government made that decision and the precise impact it had on happiness levels.

We thank the reviewer for these comments. We have expanded on this section (please see lines 557 – 571 on page 23) to explain the New Zealand government's COVID-19 strategy of 'going hard and fast'. Additionally, we elaborated to strengthen our argument that these lockdowns should not have happened given the high cost to both the economy and New Zealanders' happiness.

---

## [Editor Report · Decision Letter 1]

22 Oct 2021

The evolution of happiness pre and peri-COVID-19: a Markov Switching Dynamic Regression Model.

PONE-D-21-12326R1

Dear Dr. Rossouw,

We’re pleased to inform you that your manuscript has been judged scientifically suitable for publication and will be formally accepted for publication once it meets all outstanding technical requirements.

Kind regards,

Maximo Rossi, PhD Economics

Academic Editor

PLOS ONE

Additional Editor Comments:

Dear Stephanie, I consider that you have adequately taken into account the recommendations that we have made. In the case that you maintained some aspects, you have justified the decision and I consider it appropriate as well. In my opinion the paper is novel, it has been well exposed and therefore has the qualities to be published. Best. Maximo
---

## [Editor Report · Acceptance letter]

3 Dec 2021

PONE-D-21-12326R1 

The evolution of happiness pre and peri-COVID-19: a Markov Switching Dynamic Regression Model. 

Dear Dr. Rossouw:

I'm pleased to inform you that your manuscript has been deemed suitable for publication in PLOS ONE. Congratulations! Your manuscript is now with our production department. 

Kind regards, 

on behalf of

Dr. Maximo Rossi 

Academic Editor

PLOS ONE